# A Feasibility Study for Immediate Histological Assessment of Various Skin Biopsies Using Ex Vivo Confocal Laser Scanning Microscopy

**DOI:** 10.3390/diagnostics12123030

**Published:** 2022-12-02

**Authors:** Hanna Ogrzewalla, Matthias Möhrle, Gisela Metzler, Thomas Eigentler, Anne-Kristin Münch, Stephan Forchhammer

**Affiliations:** 1Department of Dermatology, Eberhardt Karls Universität, 72074 Tübingen, Germany; 2Praxisklinik Haut und Venen, 72072 Tübingen, Germany; 3Zentrum für Dermatohistologie und Oralpathologie Tübingen/Würzburg, 72072 Tübingen, Germany; 4Department of Dermatology, Venereology and Allergology, Charité—Universitätsmedizin Berlin, Corporate Member of Freie Universität Berlin and Humboldt-Universität zu Berlin, Luisenstrasse 2, 10177 Berlin, Germany; 5Institut für Klinische Epidemiologie und angewandte Biometrie, Eberhardt Karls Universität, 72074 Tübingen, Germany

**Keywords:** CLSM, dermatopathology, inflammatory diseases, epithelial tumors, melanocytic tumors

## Abstract

Background: Digitally stained ex vivo confocal laser scanning microscopy (CLSM) scans are a possible alternative to formalin-fixed and paraffin-embedded (FFPE) and hematoxylin-eosin (H&E) stained slides. This study explores the diagnostic accuracy of digitally-stained CLSM scans in comparison to H&E-stained slides in various dermatologic diseases in a real-life setting. Methods: Samples of patients out of one selected dermatologic office were primarily scanned via CLSM; a diagnosis was made afterwards using FFPE- and H&E-stained slides by two experienced dermatopathologists. Primary outcomes were sensitivity and specificity of diagnosis in digitally stained CLSM scans in three separate diagnostic groups. Results: CLSM evaluation of epithelial tumors (n = 132) demonstrated a sensitivity of 64.3%/83.9% and a specificity of 84.2%/71.1%. Diagnosis of melanocytic tumors (n = 86) showed a sensitivity of 19.1%/85.1% and a specificity of 96.3%/66.7%. In the diagnosis of other tumors/cysts and inflammatory dermatoses (n = 42), a sensitivity of 96.4%/96.8% and a specificity of 57.1%/45.5% was reached. Conclusions: This study shows the possibilities and limitations of a broad use of CLSM. Because of a partly low diagnostic accuracy, such an application does not seem to be recommendable at present for every indication.

## 1. Introduction

The diagnosis of formalin-fixed and paraffin-embedded (FFPE), as well as hematoxylin-eosin (HE) stained slides, is the gold standard in the diagnosis of numerous inflammatory dermatological diseases as well as tumors. This technique is established worldwide and offers numerous advantages, such as the possibility of long-term archiving or the possibility of immunohistochemical processing with various antibodies. In addition, the diagnostic criteria of numerous diseases are optimized for the specific morphology of FFPE embedded and H&E-stained section preparations. Nevertheless, there is a crucial disadvantage of the technique: due to the necessary fixation, sectioning, embedding, and staining, the method is very time consuming. In a routine clinical setting, it takes approximately from 20 to 24 h from the time of specimen collection to the creation of the H&E-stained section preparation. Furthermore, it is possible to examine excised tissue using cryostat sections, but this is very resource intensive.

With the introduction of ex vivo confocal laser scanning microscopy (CLSM), a possible alternative has been available for several years. It is now possible to scan fresh, unfixed tissue and create digital histological sections from them within minutes [1]. These sections can be created in reflection mode or fluorescence mode, but also combined in the so-called fusion mode. The latest generation of CLSM scanners and software makes it possible to digitally stain these primarily black and white images. This allows the creation of pink- and blue-dyed sections which are similar to the conventional H&E staining [2,3,4]. The familiar staining pattern should allow dermatopathologists to reliably diagnose CLSM sections without a long familiarization. Another possible application of CLSM, especially in dermatology, lies in the imaging of living tissue samples (in vivo CLSM). Here, non-invasive and dynamic images can be obtained, especially of easily accessible organ systems such as the eye and skin, which in their resolution now comes close to light microscopic imaging [5,6,7].

There are several studies investigating the use of ex vivo CLSM in the evaluation of dermatological indications. Most of the data are available for the diagnosis of basal cell carcinomas (BCC). In this case, a high specificity and sensitivity could be demonstrated in comparison to H&E diagnostics [8,9,10,11,12,13,14,15,16,17,18,19,20,21,22,23]. An important application is particularly in the use of Mohs surgery. Here, imaging by ex vivo CLSM is a highly promising alternative to workup using cryostat sections [24]. However, reliable data on the use of CLSM in other tumor entities and inflammatory dermatoses are lacking. There are only a few studies and reports, for example for the diagnosis of squamous cell carcinoma or atypical fibroxanthoma, that address the evaluation of these entities [19,22,25,26].

In this study, the reporting of ex vivo CLSM sections compared with H&E-stained section preparations was investigated in a clinical routine setting. To ensure this, all excision specimens and biopsies taken in a dermatological office during the period of the study were processed using CLSM and subsequently fixed in formalin and H&E-stained. This was performed independently of the clinical diagnosis, so all skin tumors and all inflammatory dermatoses that required bioptic confirmation were included in the study. Because there are already numerous studies on the use of CLSM in the diagnosis of BCC and because this study is intended to investigate the diagnostic quality in other diseases, patients in whom the histological workup resulted in the diagnosis of BCC were evaluated in a separate evaluation with a modified setting. As many of the histological changes investigated in this paper do not yet have established criteria for CLSM reporting, experts in dermatopatholgy were chosen to report the sections instead of experts in the field of CLSM reporting. Thus, it should be explored to what extent morphological criteria which are established in H&E diagnosis can be transferred to the assessment of CLSM scans.

## 2. Materials and Methods

### 2.1. Study Design

All patients who underwent excision, shave excision, or punch biopsy at the “Praxisklinik Tübingen—Haut und Venen” from 6 April 2020 to 27 May 2020 and gave their informed consent were included in the study. The CLSM scanner “Vivascope 2500” was provided by the company Mavig GmbH, Munich, to perform the study. The indication for excision as well as the surgery itself was performed by dermatologists of the “Praxisklinik Tübingen—Haut und Venen”. Patients with histological confirmation of BCC were included in a separate study. Patients were divided into the three groups of non-BCC epithelial tumors, melanocytic tumors, and inflammatory diseases/cysts/other tumors, based on their diagnosis.

### 2.2. Preparation of CSLM Scans and H&E Sections

The macroscopic dissection of tumor excites was performed by the operating dermatologist. The preparations were cut either in bread loaf technique, in muffin-technique, or as 3D histology according to the “Tübinger Torte” [27]. Punch preparations and shave excisions were cut into representative sections. Depending on the size of the specimens, tumor center sections and tumor margins were processed into up to 7 lamellae, each of which was further evaluated as a separate specimen. The tissue samples were scanned after they were excised. If the samples could not be scanned immediately, they were stored on moist compresses to keep them from drying out. To prepare the scans, the specimens were first stained with acridine orange. They were stained for 20 s in 2.5 mL Ringer-lactate solution (Braun) with 20 drops of acridine orange. After staining, the specimens were scanned with the VivaScope 2500, and digital scans were made. Afterward, the tissue was placed in cassettes and fixed in formalin. H&E sections were then prepared from this without further macroscopic dissection.

### 2.3. Findings

The reporting of the CLSM sections, as well as the H&E sections, was performed by two dermatopathologists (GM, SF). Both are experienced in the diagnosis of H&E-stained slides; there was no previous experience in the diagnosis of CLSM scans. Both pathologists were trained in the reporting of CLSM sections, and the sections from 6 patients were made available for this purpose as both conventional H&E histology and CLSM scans. For reporting, the site, age, and suspected clinical diagnosis were provided. Reference findings were made on the H&E sections without resorting to further immunostaining. In case of discrepancies in the diagnosis of H&E-based slides by both pathologists, the sections were microscoped together and a consensus was formed. The CLSM findings were performed at an 8-week interval from the H&E findings. This was done using the software VivaScan provided by Mavig. Each CLSM scan was assessed for quality of scan (1–6 with 1 being the best and 6 the worst quality), time of evaluation, and percentage of epidermis present.

### 2.4. Statistics

Data were evaluated using SPSS, IBM (Version 27). Numerical variables were described by mean value and standard deviation or median values and interquartile range (IQR). Because a part of the data was not normally distributed, only the median was calculated for most of the data. The number of false-positive and false-negative diagnoses was recorded. On the basis of this, the sensitivity and specificity were calculated. Sensitivity and specificity were calculated separately for the two dermatopathologists (DP1/DP2).

## 3. Results

Samples were collected from 55 patients. Because many of these patients received multiple excisions, a total of 66 samples were collected. After tissue processing, a total of 260 specimens were obtained which were histologically diagnosed by ex vivo CLSM as well as by the H&E section. The epidemiological data of the study collective and the specimen locations are shown in Table 1.

As numerous tumor margins were present in the study cohort, the most common diagnosis was “normal skin/no tumor”. The breakdown of the diagnostic groups mentioned in the study is shown in Table 2. In the following, we divided the evaluation into three major groups: epithelial tumors (without BCC), melanocytic tumors, and as a third group, inflammatory diseases as well as further tumors.

### 3.1. Epithelial Tumors

In the non-BCC epithelial tumor group, 132 specimens from a total of 23 patients were examined. The group included specimens with seborrheic keratoses (see Figure 1a,b), trichoepitheliomas, solar keratoses, and Bowen’s disease (see Figure 1c,d) as well as squamous cell carcinomas (highly differentiated and desmoplastic) (see Figure 1e,f; Appendix A). The sensitivity was 64.3% (DP1)/83.9% (DP2) and the specificity 94.2% (DP1)/71.1% (DP2), respectively. Overall, 100 (DP1)/101 (DP2) of the 132 specimens were correctly diagnosed. Of the 132 specimens in this group, 40 excites were processed as so-called “muffin”, and 55 tumor center sections, 24 tumor margins, 5 tumor bases, and 8 punch biopsies were available. The median time of diagnosis was 20 (DP1)/45 (DP2) seconds, the specimens were evaluated with a median quality of 2 (DP1/DP2), and the epidermis showed a median of 90% (DP1)/80% (DP2). The median time for creating a digital CLSM scan was 5.75 min.

### 3.2. Melanocytic Tumors

In the group of melanocytic tumors, 86 sections of 21 patients were found. The group included specimens with melanocytic nevi (junctional, compound, and dermal nevi) (see Figure 2a,b), dysplastic melanocytic nevi (junctional and compound nevi), and lentigo maligna (melanoma in situ) (see Figure 2c,d, Appendix A). Here, only a sensitivity of 19.1% (DP1)/85.1% (DP2) and a specificity of 96.3% (DP1)/66.7% (DP2) were found. There were 34 (DP1)/66 (DP2) of the 86 specimens correctly diagnosed, respectively. In the group of melanocytic tumors, 17 specimens were processed as “muffin”, and 64 specimens showed tumor center sections; furthermore, 5 section margins were available for reporting. The median time of diagnosis was 20 (DP1)/60 (DP2) seconds, the sections were evaluated with a median quality of 2 (DP1)/3 (DP2), the epidermis showed a median of 90% (DP1)/80% (DP2).

### 3.3. Inflammatory Diseases, Cysts, and Other Tumors

A total of 42 specimens from 14 patients were classified in the group of inflammatory dermatoses, cysts, and other tumors. Specimens of lichen planus (see Figure 3a,b, Appendix A), chronic eczema and unspecific lymphocytic infiltrates, epidermal cysts (see Figure 3c,d, Appendix A), and abscessing inflammatory reactions, as well as dermatofibromas (see Figure 3e,f, Appendix A), angiofibromas, fibromatoses, and lipomas were found. This group showed a diagnostic accuracy of 96.4% (DP1)/96.8% (DP2) for sensitivity and 57.1% (DP1)/45.5% (DP2) for specificity, respectively. Both dermatopathologists correctly diagnosed 35 of the 42 specimens. There were 1 muffin, 18 punch biopsies, and 23 tumor centers. The median time of diagnosis was 15 (DP1)/50 (DP2) seconds, the sections were evaluated with a median quality of 2 (DP1/DP2), and the epidermis showed a median of 100% (DP1)/90% (DP2).

## 4. Discussion

Although the study cannot provide an entire picture of the reporting of CLSM sections due to the overall, rather small number of different diagnoses, it does show a representative result from the real-life setting of a dermatology practice performing surgery. Sensitivity and specificity in this study are shown to be dependent on both the examiner and the entity being examined.

Partially, there was a discrepancy in diagnostic accuracy between the two dermatopathologists. Neither of the dermatopathologists had prior clinical experience in reporting CLSM scans; however, both received specialized training in the diagnosis of CLSM scans. Previous work reported that it can be possible for untrained examiners to diagnose CLSM scans [28]. However, the appearance of melanin-containing structures is especially different from the appearance in H&E-stained FFPE slides [29]. This limits the ability to directly transfer knowledge about the evaluation of H&E-stained FFPE slides to the digitally-stained CLSM scans.

It still can be possible for inexperienced investigators to directly evaluate CLSM scans without training. However, given the very heterogeneous results in this study, it seems reasonable to also offer intensive training to experienced dermatopathologists before they evaluate digitally stained CLSM scans.

CLSM scans of epithelial tumors showed quite good results of sensitivity and specificity compared to findings of H&E-stained sections. An advantage in this group is that many structures present similarly in CLSM scan as in H&E staining [22,26]. This facilitates the diagnosis by pathologists trained in H&E sectioning and leads to fewer misdiagnoses in our study. High diagnostic accuracy in the diagnosis of SCC using ex vivo CLSM has also been found in other studies [23,26].

The reporting of melanocytic tumors caused the greatest problems in our study. This is reflected in a significantly poorer diagnostic accuracy in this group. Although it was relatively easy to diagnose the dermal, mostly less pigmented parts of melanocytic nevi, there were major problems, especially in the evaluation of the junctional tumor component. This made it almost impossible to evaluate marginal controls of a lentigo maligna. A major problem here could be the different presentation of melanin pigment in CLSM compared to H&E findings. Thus, melanin pigment, whether located in keratinocytes or melanocytes, shows an intense pink staining signal in the CLSM scan which hardly allows for a morphological evaluation (see Figure 2c) [3]. This makes the detection of confluent melanocytes and melanocyte nests, as well as a morphological characterization of the melanocytic cell component, almost impossible. Possibly, a longer familiarization with CLSM sections of melanocytic tumors could also improve the diagnostic accuracy, after all, the diagnosis of these tumors is partly one of the most challenging tasks in dermatopathology even in the H&E section. In addition, it must be mentioned that the spectrum of melanocytic tumors investigated in our study is quite small, i.e., no invasive melanoma was available for diagnosis. Nevertheless, our study shows the potential possibility but also the pitfalls of such an application. Further studies investigating the diagnostic accuracy of melanocytic tumors are not available to our best knowledge.

In the group of inflammatory skin diseases, cysts, and other tumors, results for sensitivity were relatively high. These high values are surprising given a large number of possible different diagnoses in this group. But as other studies have previously shown, the majority of inflammatory skin diseases (psoriasis, eczema, lichen planus, and lupus erythematosus) can be well visualized with ex vivo CLSM [30]. The lymphocytic infiltrates as well as neutrophilic granulocytes are well distinguishable [31]. Although a rapid diagnosis of the inflammatory pattern seems to be possible via ex vivo CLSM, conventional histology should always be performed to make a final diagnosis.

The biggest limitation of our study is certainly the small number of cases. Although a large proportion of samples from a dermatologic practice doing surgery were processed over a period of about 7 weeks, only limited cases of individual diagnoses could be collected. In order to obtain more reliable statistical results, it would be necessary to investigate individual diagnostic groups over a longer period of time and with a larger number of cases. Another limitation is that the investigators had no previous clinical experience in the diagnosis of ex vivo CLSM. However, because the study was intended to investigate all dermatological diseases, some of which have not yet been described in ex vivo CLSM diagnostics, this approach was chosen in order to transfer the morphological knowledge of the H&E findings to the digitally generated image of CLSM.

## 5. Conclusions

In summary, this feasibility study of ex vivo CLSM in a dermatologic practice doing surgery shows the possibilities and some limitations of a broad use of CLSM. A large proportion of the diagnoses that are examined in a routine setting can be diagnosed with quite a high accuracy via CLSM. The usage of ex vivo CLSM may be feasible in the clinical setting, especially when rapid detection of epithelial tumors or inflammatory skin lesions is required. At present, an implication does not seem to be recommendable for melanocytic lesions due to partly low diagnostic accuracy. Given the partially high values in diagnostic accuracy, an application of CLSM for certain tumor entities seems conceivable; however, larger studies are needed to prove this.

## Figures and Tables

**Figure 1 diagnostics-12-03030-f001:**
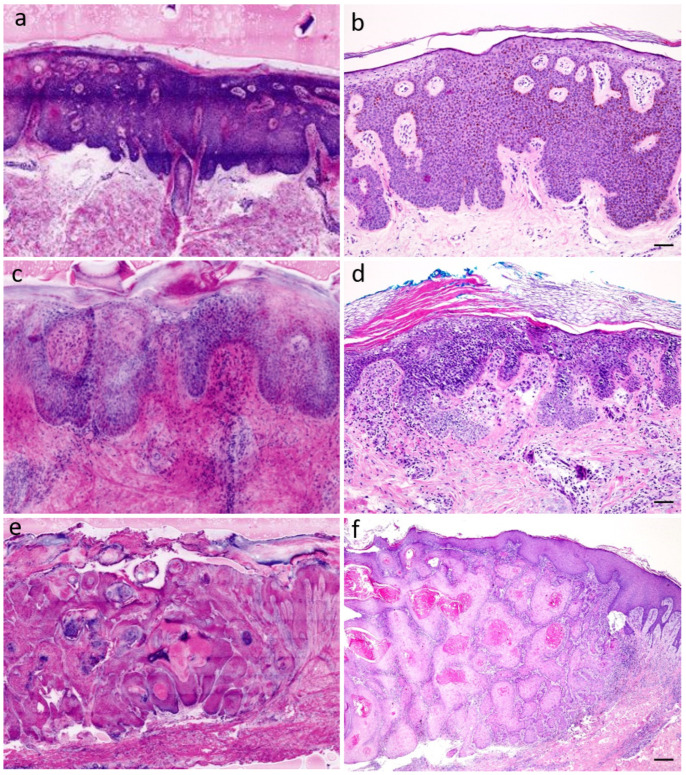
Epithelial tumors (**a**) seborrheic keratosis, CLSM (**b**) seborrheic keratosis, H&E-stain, scale = 250 µm. (**c**) Bowen’s disease (squamous cell carcinoma in situ), CLSM (**d**) Bowen’s disease, H&E-stain, scale = 100 µm (**e**) highly differentiated squamous cell carcinoma (SCC, G1), CLSM (**f**) SCC, G1, H&E-stain, scale = 500 µm.

**Figure 2 diagnostics-12-03030-f002:**
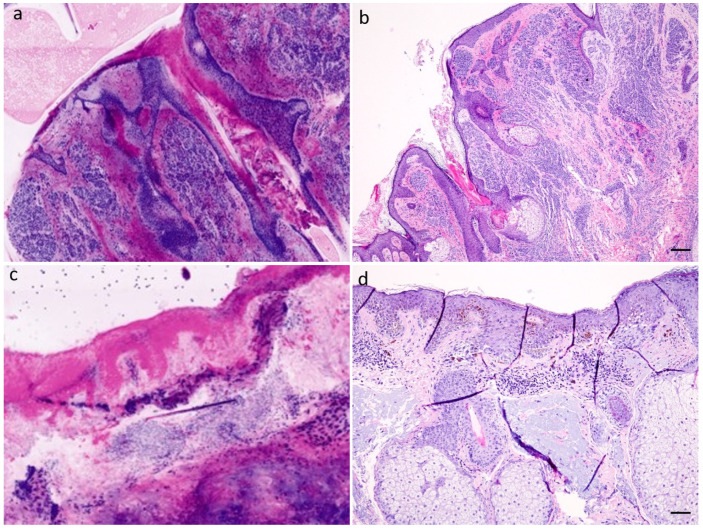
Melanocytic tumors (**a**) papillomatous dermal nevus, CLSM (**b**) papillomatous dermal nevus, H&E-stain, scale = 250 µm (**c**) lentigo maligna (melanoma in situ), CLSM; (**d**) lentigo maligna, H&E-stain, scale = 100 µm.

**Figure 3 diagnostics-12-03030-f003:**
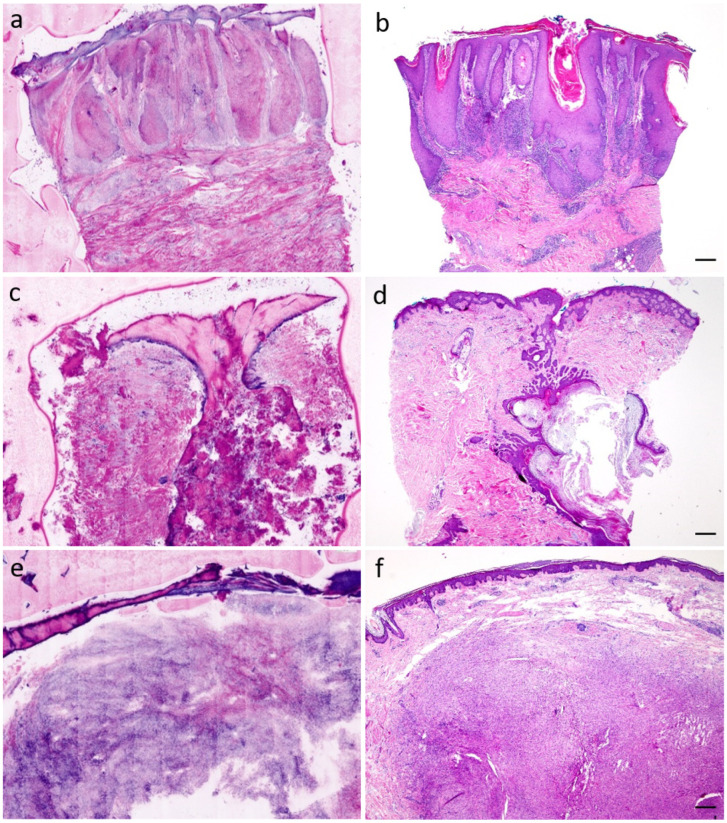
Inflammatory dermatoses, cysts, and other tumors. (**a**) lichen planus verrucosus, CLSM (**b**) lichen planus verrucosus, H&E-stain, scale = 500 µm (**c**) epidermal cyst, CLSM (**d**) epidermal cyst, H&E-stain, scale = 500 µm (**e**) dermatofibroma, CLSM (**f**) dermatofibroma, H&E-stain, scale = 500 µm.

**Table 1 diagnostics-12-03030-t001:** Epidemiologic data and specimen localization.

**Age** (n = 55)	
Min./Max.	21/94
Median (+ IQR)	69.5 (55/80)
Mean value (±SD)	66.18 (±17.39)
**Sex** (n = 55)	
Male (n, %)	21 (38.2%)
Female (n, %)	34 (61.8%)
**Localization** (n = 260)	
Capillitium	9 (3.5%)
Forehead	39 (15%)
Eyebrow	9 (3.5%)
Eye/Eyelid	12 (4.6%)
Nose	18 (6.9%)
Ear	12 (4.6%)
Cheek	29 (11.2%)
Philtrum	1 (0.4%)
Neck	5 (1.9%)
Shoulder	15 (5.8%)
Chest	6 (2.3%)
Arms	22 (8.5%)
Back	15 (5.8%)
Abdomen	4 (1.5%)
Buttock	13 (5%)
Upper leg	6 (2.3%)
Lower leg	36 (13.8%)
Not specified	9 (3.5%)

**Table 2 diagnostics-12-03030-t002:** Diagnostic groups.

Diagnosis	Frequency (n, %)
**Epithelial tumors**	**132 (50.8%)**
No Tumor/”normal“ skin	64 (24.6%)
Seborrheic keratosis	21 (8.1%)
Trichoepithelioma	2 (0.7%)
Solar Keratosis	4 (1.5%)
Bowen’s disease	13 (5%)
Highly differentiated SCC	18 (6.9%)
Low differentiated/desmoplastic SCC	10 (3.8%)
**Melanocytic tumors**	**86 (33.1%)**
No Tumor/”normal“ skin	28 (10.8%)
Melanocytic naevus/naevoid lentigo	26 (10%)
Dysplastic melanocytic naevus	10 (3.8%)
Lentigo maligna/Melanoma in situ	22 (8.5%)
**Inflammatory diseases, cysts, and other tumors**	**42 (16.2%)**
Lichen planus	11 (4.2%)
Prurigo/chronic eczema	4 (1.5%)
Lymphocytic infiltrate	4 (1.5%)
Abscess/Epidermal Cyst/Lipoma	14 (5.4%)
Dermatofibroma/Angiofibroma/Fibromatosis	9 (3.5%)

## Data Availability

The data presented in this study are available on request from the corresponding author.

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
