# Peer review of "A Feasibility Study for Immediate Histological Assessment of Various Skin Biopsies Using Ex Vivo Confocal Laser Scanning Microscopy"

_diagnostics, 2022, doi:10.3390/diagnostics12123030_

Round 1
Reviewer 1 Report
In this manuscript, authors have described digitally stained ex-vivo confocal laser scanning microscopy (CLSM) scans as an alternative to conventional formalin-fixed and paraffin-embedded (FFPE) and hematoxylin-eosin (H&E) stained slides. They also evaluated the diagnostic accuracy of digitally stained CLSM scans relative to H&E stained slides in various dermatologic diseases. Authors detailed the shows the potential and and the limitations of a broad use of CLSM with regard to applications. This is an interesting study and quite useful in pathology. Overall manuscript is written well and a few recommendations for the improving the manuscript is provided below.
1. Authors provided sensitivity values as, example, 64,3%. Do they mean 64.3 % ? Could authors make corrections with these decimal representations?
2. Could authors provide a quantitative way of comparing the stained images?
3. I think authors didn't enough highlight the potential of the CSLM for imaging of live tissue samples. This would be a excellent usage of CSLM for the in situ living tissue imaging with unstained samples (autofluoroscence (AF) imaging) or with stained imaging. This possible with the live tissue chamber as demonstrated in the following references[1-2]. In that case, contrast of the image comes from the AF . Authors can mention this in the text (with ref.1 in line 50) ((1.) https://doi.org/10.1167/iovs.61.13.1 (2) https://doi.org/10.1038/s41598-021-95320-z)
4. Do you have in vivo image of this samples to add into the manuscript for a quick comparison for the readers ?
Author Response
Manuscript ID: diagnostics-2040804
Response to Reviewer 1
Dear Reviewer,
Thank you for giving us the opportunity to submit a revised version of our paper "Feasibility study for immediate histological assessment of various skin biopsies using ex-vivo confocal laser scanning microscopy." We highly appreciate the time and effort you have devoted to providing feedback on our manuscript as a reviewer. We are very grateful for the insightful comments and valuable improvements to our paper. The majority of the proposed changes were incorporated into the manuscript, the alterations are marked with tracked changes in the revised manuscript. Page and line numbers in this response refer to the revised manuscript.
Reviewers comments:
In this manuscript, authors have described digitally stained ex-vivo confocal laser scanning microscopy (CLSM) scans as an alternative to conventional formalin-fixed and paraffin-embedded (FFPE) and hematoxylin-eosin (H&E) stained slides. They also evaluated the diagnostic accuracy of digitally stained CLSM scans relative to H&E stained slides in various dermatologic diseases. Authors detailed the shows the potential and the limitations of a broad use of CLSM with regard to applications. This is an interesting study and quite useful in pathology. Overall manuscript is written well and a few recommendations for the improving the manuscript is provided below.
Author response: Thank you very much!
- Authors provided sensitivity values as, example, 64,3%. Do they mean 64.3 % ? Could authors make corrections with these decimal representations?
Author response: All values were corrected accordingly.
- Could authors provide a quantitative way of comparing the stained images?
Author response: As described in the methods section, the results of the CLSM scans were measured against the reference findings of the HE morphology. Accordingly, the CLSM scan was classified as "correctly diagnosed" or "incorrectly diagnosed".
The evaluation of CLSM scan quality was performed semiquantitatively. As described in the method section, school grades from 1 to 6 were assigned, where 1 describes the best possible result and 6 describes a scan that cannot be evaluated diagnostically.
- I think authors didn't enough highlight the potential of the CSLM for imaging of live tissue samples. This would be an excellent usage of CSLM for the in situ living tissue imaging with unstained samples (autofluoroscence (AF) imaging) or with stained imaging. This possible with the live tissue chamber as demonstrated in the following references[1-2]. In that case, contrast of the image comes from the AF. Authors can mention this in the text (with ref.1 in line 50) ((1.) https://doi.org/10.1167/iovs.61.13.1 (2) https://doi.org/10.1038/s41598-021-95320-z)
Author response: The potential use of in-vivo CLSM of living tissue was included in the introductory section as suggested. The corresponding attached references have been cited and included in the literature list. (page 2, line 67-71)
- Do you have in vivo image of this samples to add into the manuscript for a quick comparison for the readers?
Author response: There are no in vivo images of the tumors or inflammatory dermatoses from the patients in the study before surgery. However, all tumors were photographed macroscopically after cutting. For reference, the macroscopic images corresponding to the selected photographs in Figures 1-3 were included in the paper as supplementary figures. (see supplementary figures S1-3; page 13, line 501-515)

Reviewer 2 Report
This is a report about the use of ex vivo confocal microscopy for the detection of cutaneous tumors. In this regard in last years ex vivo RCM showed an increase of application and therefore it's important for the readers, study articles about this topic. The paper is well written, some minor changes are needed:
- please specify that the use of ex vivo can be expanded also in different surgical areas, such as the Mohs surgery. This is important for the readers since the application of ex vivo RCM is veryb high for sirgeons that perform Mohs surgery. In this regard, please read and add to your references thi article that explain that explain the role of ex vivo RCM for Mohs surgery: "Digital ex-vivo confocal imaging for fast Mohs surgery in nonmelanoma skin cancers: An emerging technique in dermatologic surgery. Dermatol Ther. 2019 Nov;32(6):e13127. doi: 10.1111/dth.13127. Epub 2019 Nov 12. PMID: 31628777."
- when you explain that ex vivo can be used also for other tumors, please note that in litertaure there are also cases of use of ex vivo RCM for atypical fibroxanthoma. Thi could be intersting for the readers to better know the application of this new tecnique.
- I donìt think that ex vivo RCM can be valid for inflammatory cutaneous conditions; so please specify that an histological examination is always needed.
Thank you
Author Response
Manuscript ID: diagnostics-2040804
Response to Reviewer 2
Dear Reviewer,
Thank you for giving us the opportunity to submit a revised version of our paper "Feasibility study for immediate histological assessment of various skin biopsies using ex-vivo confocal laser scanning microscopy." We highly appreciate the time and effort you have devoted to providing feedback on our manuscript as a reviewer. We are very grateful for the insightful comments and valuable improvements to our paper. The proposed changes were incorporated into the manuscript, the alterations are marked with tracked changes in the revised manuscript. Page and line numbers in this response refer to the revised manuscript.
Reviewers comments:
This is a report about the use of ex vivo confocal microscopy for the detection of cutaneous tumors. In this regard in last years ex vivo RCM showed an increase of application and therefore it's important for the readers, study articles about this topic. The paper is well written, some minor changes are needed:
Author response: Thank you very much!
- please specify that the use of ex vivo can be expanded also in different surgical areas, such as the Mohs surgery. This is important for the readers since the application of ex vivo RCM is very high for surgeons that perform Mohs surgery. In this regard, please read and add to your references the article that explain the role of ex vivo RCM for Mohs surgery: "Digital ex-vivo confocal imaging for fast Mohs surgery in nonmelanoma skin cancers: An emerging technique in dermatologic surgery. Dermatol Ther. 2019 Nov;32(6):e13127. doi: 10.1111/dth.13127. Epub 2019 Nov 12. PMID: 31628777."
Author response: The use of ex vivo particularly in the area of Mohs surgery has been added as recommended and included in the introduction. The corresponding references have been inserted and added to the literature list. (page 2, line 75-77)
- when you explain that ex vivo can be used also for other tumors, please note that in literature there are also cases of use of ex vivo RCM for atypical fibroxanthoma. This could be interesting for the readers to better know the application of this new technique.
Author response: Thank you for this remark! A corresponding literature reference and a short note have been added in the introductory section.(page 2, line 79-80)
Pampena, R., Peccerillo, F., Piana, S., Paolino, G., Mercuri, S. R., Pellacani, G., & Longo, C. (2019). Atypical fibroxanthoma: in-vivo and ex-vivo confocal features. Giornale Italiano di Dermatologia e Venereologia. https://doi.org/10.23736/S0392-0488.19.06319-3
- I don’t think that ex vivo RCM can be valid for inflammatory cutaneous conditions; so please specify that a histological examination is always needed.
Author response: We fully agree with your statement. A corresponding restriction on the use of CLSM for inflammatory skin diseases has been added in the discussion section. (page 9, line 312-314)
